# Little Owl Aggression and Territory in Urban and Rural Landscapes

**DOI:** 10.3390/ani14020267

**Published:** 2024-01-15

**Authors:** Grzegorz Grzywaczewski, Federico Morelli, Piotr Skórka

**Affiliations:** 1Department of Zoology and Animal Ecology, University of Life Sciences in Lublin, PL 20-950 Lublin, Poland; 2Institute of Biological Sciences, University of Zielona Góra, Prof. Szafrana St. 1, PL 65-516 Zielona Góra, Poland; fmorellius@gmail.com; 3Faculty of Environmental Sciences, Czech University of Life Sciences Prague, 165 00 Prague, Czech Republic; 4Institute of Nature Conservation, Polish Academy of Sciences, PL 31-120 Kraków, Poland; skorasp@gmail.com

**Keywords:** territory size, urbanization, habitat composition, Minimum Convex Polygon

## Abstract

**Simple Summary:**

In this study, we explored the territorial displays of Little Owls in urban and rural areas. Specifically, we compared the size of territories between urban and rural landscapes, investigating the main characteristics (e.g., land use composition, altitude) that characterize territories of the species in both types of landscapes. Overall, territories were smaller in urban than in rural areas, with a relatively different land use composition. Finally, even though the rate of territorial displays was similar between urban and rural territories, individuals used various types of structures differentially. Our findings provide new insight concerning the conservation of Little Owls in anthropized areas.

**Abstract:**

Urbanization is a major land use change across the globe with vast effects on wildlife. In this paper, we studied (1) the territorial displays of Little Owls in urban and rural landscapes, analyzing also (2) the size and habitat composition of the territories, and (3) the factors affecting territory size in both landscapes. To do that, we used *t*-tests, Principal Components Analysis, and General Linear mixed model procedures. The territory size was smaller in urban than in rural landscapes. Urban territories of Little Owls are characterized by a lower cover of grassland, tall crops, short crops, gardens, and orchards, as well as a higher cover of built-up areas than territories in rural landscapes. Territory size in rural landscapes was negatively correlated with seasonal progress and positively correlated with altitude. The rate of territorial displays was similar between urban and rural territories; however, birds differentially utilized various structures. In urban territories, birds mostly used buildings, whereas in rural territories, birds used electric pylons and trees. The compositional differences between territories in the two landscapes may have important consequences for other behavior types and possibly reproductive output in this species.

## 1. Introduction

Urbanization is the fastest-growing land use worldwide, with half of the human population residing in urban areas [1,2,3]. Alongside humans, many animal species and plants, previously associated with more natural environments like rural landscapes, have started to occur in towns (e.g., [1,4,5]). Urban areas substantially differ from other landscapes; therefore, to be colonized by species, they must contain basic resources enabling survival and reproduction [5,6,7]. Also, the successful urban colonizer must adopt or express substantial behavioral plasticity to different habitat structures and the density of the human population in towns [5,8,9]. Although many studies have analyzed the urbanization process, much less is understood about how animal populations differ in habitat requirements between urban and rural areas and how this difference mediates the behavior of animals [8,10,11,12,13,14]. The selection of sites by animals to meet their requirements is a behavioral process itself; however, it is more related to physical components of the environment rather than any other types of behavior linked with the presence of conspecifics or different species (e.g., aggressive behavior or flocking). Thus, to understand the urbanization process, habitat requirements and other types of behavior related to social context should be considered separately [15]. Basically, four scenarios can predict how urbanization may affect habitat requirements and the behavior of animals. First, individuals may have the same habitat requirements in urban and rural environments, resulting in similar social behavior and occurrence in areas with similar features or equivalents [16]. Second, it is also possible that species in urban and rural environments may have similar habitat requirements, but their behavior may differ [17]. Third, animals in urban environments may have different habitat requirements than in rural landscapes to adjust to the elevated human abundance, car traffic, foraging areas, etc. [18]. Thus, the behavior should mirror the difference between urban and rural landscapes. Finally, animals in urban environments may have different habitat requirements than in rural landscapes, but their behavior may stay unchanged across environments ([19], but see: [10]). The first scenario indicates that urbanization is purely a matter of animal selection of sites similar to those in more rural areas with respect to physical composition. In the second case, urbanization is a process where habitat requirements remain constant, but at least plasticity of other behaviors is involved. In the third scenario, urbanization involves both changes in habitat requirements and behavior. In the fourth scenario, urbanization is a process driving the habitat requirements but not a social behavior. In addition, other studies have shown that some species of owls have an extremely high level of personal fear of people, which affects the lifespan of these birds [20].

The Little Owl *Athene noctua* in Europe occurs in open landscapes, especially in areas that have been anthropogenically transformed and covered with low vegetation, in both rural and urban landscapes. Its original habitats vary from sandy dunes, steppes, and stony semi-deserts to agricultural land with hedgerows and trees, and woodland verges. As hunting places, it prefers fields, meadows, pastures, gardens, and built-up areas, particularly roads, tree belts, fences, edges of fields, and meadows (e.g., [21,22,23,24,25,26,27,28,29,30,31,32,33,34,35,36,37]). The urban landscape was colonized by this species as the last one, following river valleys and farmland [38,39,40,41]. However, only during the last 20–30 years has more attention been paid to this phenomenon, with most studies reporting only population density and characteristics of the sites in towns [39,40,42,43,44,45,46,47]. However, differences in habitat structures used by individuals are mirrored in behavioral differences in Little Owls [48].

Similarly to other owls (and many other birds), the Little Owl is strictly territorial and exhibits specific territorial behavior [23,49,50]. The size of the territory varies throughout the year and among years [23,49,50]. It may differ due to variable weather conditions and geographical locations. However, nowadays there is no data on the differences between habitat requirements and territorial behavior of Little Owls in urban and rural environments [39,40]. This knowledge is necessary to understand wider habitat-related population trends of this species. In Central and Eastern Europe, this species is stable in urban environments [45,46], while in rural areas, the occurrence of the species decreases both in site occupancy and abundance [26,28,51,52,53].

This paper aims to study how some characteristics of the breeding ecology of the Little Owl differ between urban and rural landscapes, focusing mainly on (1) territorial aggressive displays, (2) territory size and habitat composition within territories, and (3) factors that affect territory size.

## 2. Materials and Methods

### 2.1. Study Area

The study was carried out in the central part of the species’ geographical range—in the Lublin Region, Eastern Poland (Figure 1). This is a typical agricultural region with over 50% of its area covered by fertile soil (mostly in the upland areas). The bedrock is loess and sand. In total, arable land covers 69% of the region [54]. Human population density is low at 89 persons per 1 km^2^. Orchards are common elements of the agricultural landscape in this region. Forests cover 22% of the area. The industrial infrastructure is low, and, in effect, the urbanization level is low. There are 43 towns with a median size of 9640 inhabitants. The largest town is Lublin (350,462 inhabitants), and the smallest town is Frampol (1408 inhabitants). 

The climate in the Lublin Region is continental. The mean annual temperature is 7.0–7.5 °C [54]. Similarly to the rest of Poland, the warmest months are July and August, with mean temperatures varying between 16.5 and 17.6 °C. Summer lasts 93–97 days. The coldest month is February, and winter usually starts at the end of November in the north-eastern part of the region. The snow cover lays for 60–100 days [54].

The vegetation period, with mean daily temperatures over 5 °C, usually extends for 208–218 days. The region has thermal conditions that are of average suitability for agriculture in Poland. Precipitation is 550–700 mm [54,55].

### 2.2. Aggressive Displays, Territory Size, and Land Cover Composition

The study was carried out from 2002 to 2004, between 15 April and 12 August. Field surveys were done during windless nights with no rainfall, usually between sunset and midnight when the vocal activity of Little Owls is its highest [21,23]. 

Most nocturnal owls respond to broadcasts of conspecific recordings, and this technique can be used to study their behavior [56,57] and map territories [23,58]. As Little Owls make extensive use of acoustic communication [59], to establish the boundaries of territories, we modified the above-mentioned method of Finck [23]. Finck [23] also defines the territory as the defended area for food and nesting. We introduced modifications to this method to achieve better efficiency of field surveys. We used a soft plastic dummy owl that was set on top of a 2 m tall pole. The dummy owl was attached to the pole in such a manner that a real owl attacking it could not be injured. When a real owl attacked the dummy, the latter simply dropped. The observer was hidden 10–20 m from the pole with a dummy owl, and played back the territorial voice of the Little Owl. All observations were mapped in detail, and every observation point was separated by a distance of 100–200 m, as observers moved in all directions. The last point where attacks were noted was regarded as the boundary of the territory. During the night, control attacks were noted at 378 points (range 1–26, mean 6.3).

Other signs of territorial defence used to establish territorial boundaries were: (1) pretended attacks toward a dummy (where a real owl flew low over the dummy but physical contact was absent), and (2) the presence of sites where birds uttered territorial calls—the time that was required to establish the boundaries of the territory varied between 30 min and 5.5 h. The Minimum Convex Polygon method was used to establish boundaries of owls’ territories. The area of territories and the share of land cover types within them were calculated using ArcGIS 10 [60]. During observations, we noted all kinds of sites (low and tall buildings, trees, poles, etc.) from which birds uttered voices to defend their territories. In each territory, we mapped all land cover types and types of crops, and we measured the height of vegetation during observations (Figure 2). 

The sampled territories were classified as ‘urban’ or ‘rural’ following these criteria: “urban” territories were within the administrative borders of towns (largely built areas—blocks of flats, houses, communication routes, and lawns, sports fields, wastelands), and “rural” territories were within the administrative borders of villages (areas used for agriculture—fields, meadows, pastures, and houses, barns, cow-sheds, gardens, orchards), *sensu* Marzluff et al. [4]. We sampled territories in six towns and 14 villages. In total, during the study, we gathered information on 33 territories: 16 in rural landscapes and 17 in urban ones. 

### 2.3. Data Processing and Statistical Analysis

Before conducting any statistical analysis, we log-transformed almost all variables (except seasonal progress and altitude a.s.l.) because their distribution was right-skewed and to avoid the effects of detached observations [61]. Because several territories were surveyed more than once, we randomly selected one survey for each territory, representing it in the statistical analysis (with an exception, see below).

First, we tested if territory size was spatially autocorrelated by the analysis of Moran I-statistics and spatial autocorrelograms [62]. In this analysis, we tried various numbers of distance classes (from 4 to 7) with an equal number of observations to find spatial patterns. As we did not find any spatial autocorrelations (see Section 3), we used ordinary statistics in further data analysis. To compare the territory size and habitat composition within the territories of Little Owls in urban and rural landscapes, we used a *t*-test for independent samples. To find out how territory size relates to habitat variables, seasonal progress, and altitude, we used Principal Components Analysis (PCA) and the General Linear mixed model (GLMM) procedure. 

The main environmental variables that affect territory size in Little Owls were selected through PCA analysis. We used this method because most of the habitat variables were correlated with each other, and PCA is one of the multivariate methods enabling us to explore relationships among many correlated variables. We performed PCA separately for urban and rural landscapes. We allowed PCA for the first three principal components only, because further principal components did not contribute to the total variance explanation (variance explained was below 10%). We also separately tested the effect of seasonal progress on territory size, including repeated surveys within the territories. Second, GLMM was applied using territory size as the response variable, while territory number (equivalent to individual in our study) was assigned as a random effect. The use of territory identity as a random factor in models allowed for the avoidance of any pseudo-replication issues related to the high fidelity of Little Owls to breeding territories. The environmental variables described above were used as predictors. 

The G-test was used to compare the frequency of territorial activities from various elements within the territory in urban and rural landscapes. Aggressive behavior within a territory was represented by several observations (mean ± SEM: 11.4 ± 0.8 aggressive displays per territory). Because the results of this analysis may be biased due to data dependence and should be treated with caution, we also tested if the mean number of aggressive displays per hour per individual performed from various sites within the territory differed between landscapes.

Means are reported with standard errors (SEMs). All analyses were done using SAM v. 4.0 [63] and Statistica v. 10 [64] software.

## 3. Results

### 3.1. Territory Defence Sites

We found that birds defended their territories from various sites, but with different frequencies in urban and rural landscapes (G = 22.5, *p* = 0.001, Figure 3). In urban landscapes, birds defended their territories from buildings more often than in rural landscapes (Figure 3). In rural landscapes, birds defended their territories from electric poles and trees more often than in towns (Figure 3). 

The mean number of aggressive displays per hour from blocks of flats and low buildings in urban environments was significantly higher than in rural landscapes, while the number of aggressive displays from electric poles was significantly higher in rural landscapes (Table 1). However, the total number of aggressive displays did not differ between the landscapes (Table 1).

### 3.2. General Characteristics of the Territories in Urban and Rural Landscapes

The mean territory size of the Little Owl in the urban landscape was significantly smaller than the mean territory size in the rural landscape (urban: 2.15 ± 0.89 ha, range: 0.3–27.9 ha, rural: 6.08 ± 1.29 ha, range: 0.5–92.4 ha, *t* = 3.997, *df* = 49.79, *p* < 0.001). We did not find any spatial autocorrelation of territory size for both urban (Moran I = 0.014, *p* = 0.409) and rural landscapes (Moran I = −0.026, *p* = 0.308). 

Territories of the Little Owl in urban and rural landscapes showed differences in the composition of land cover typologies (Table 2). However, green lanes, roads, and wooded areas were similar in both landscape types. 

### 3.3. Factors Affecting Territory Size

The first three principal components explained 29%, 22%, 15%, and 34%, 19%, 15% of variance in territory composition in urban and rural landscapes, respectively (Figure 4). 

In urban landscapes, the first principal component represented a gradient from built-up areas to sites with a high cover of green lanes, gardens, and orchards, tall crops, and a high density of roads (Table 3). The second principal component in this landscape represented a gradient of increasing cover of grasslands and short crops (Table 3). The third principal component was represented by an environmental gradient of decreasing altitude and increasing progress of the season (Table 3). 

In rural landscapes, the first principal component represented a gradient of increasing cover of green areas: crops (both tall and short), grassland, and gardens but decreasing cover of built-up areas and green lanes (Table 3). The second principal component in this landscape represented mostly a gradient of the increasing seasonal progress and wooded areas (Table 3). The third principal component was mostly associated with decreasing altitude and increasing cover of water (Table 3).

In urban landscapes, territory size was positively correlated with the first principal component and negatively with the second one (Table 3, Figure 4).

Territory size in urban landscapes was positively correlated with the cover of tall crops, green lanes, and roads and negatively with built-up areas. The second principal component showed that territory size was negatively correlated with the cover of grassland and short crops (Table 3, Figure 4). 

In rural landscapes, territory size was highly correlated only with the second and moderately with the third principal component. Territory size in this landscape decreased with the seasonal progress and cover of water but increased with the altitude a.s.l. (Table 3, Figure 4 and Figure 5). 

The GLMM showed that the negative relationship between seasonal progress and territory size remained significant when we included repeated counts within territories (slope = −0.015 ± 0.004, GLMM F_1,49.6_ = 10.497, *p* = 0.002), but the interaction between seasonal progress and landscape type was non-significant (slope = −0.006 ± 0.004, GLMM F_1,49.6_ = 1.869, *p* = 0.178).

## 4. Discussion

Our study provides the first data on the differences in territorial aggressive behavior of the Little Owl between urban and rural landscapes. The rate of aggressive displays was similar between the two types of landscapes, suggesting that, generally, territory defence intensity in this species is similar along a gradient of urbanization. However, we also demonstrated that in the two landscapes, birds differentially utilized various structures to display this behavior. The differences corresponded to the structural components of the urban and rural territories, and birds used mostly elevated constructions during aggressive displays. In urban territories, birds commonly used buildings, whereas in rural territories, birds used electric poles and trees willingly. 

Territory size in the Little Owl in the urban environment was lower than in the rural one. This is interesting because territory size may indicate the amount of resources, with lower territories indicating a higher supply of resources [65,66]. This may suggest that the Little Owl can exploit urban environments, even though some studies have highlighted how urbanized habitat, near roads, could represent lower-quality territories for different owl species because traffic noise, in particular, may affect intra-specific communication and hunting efficiency [67]. However, lower territory size may also be a result of other potential causes, for example, a higher density of territories of owls in towns. This was not the case as we showed the rate of territorial defence was similar between the two types of environments. 

Our findings highlighted how different factors affected Little Owl territory size in urban and rural environments. In urban environments, lower territory size was associated mostly with higher covers of buildings, grasslands, and short crops, with the latter two considered typical foraging microhabitats for this species [23,68,69]. In this study, a negative correlation between territory size and the cover of buildings was found. This correlation is unexpected considering that built-up areas provide nesting sites for this species, and other studies in Central Poland have shown contrary trends [28]. At the same time, it confirms that the Little Owl needs undeveloped, open areas for hunting [39,40]. On the other hand, it is an important obstacle or barrier to inhabiting city centers with dense buildings. Moreover, food resources, namely rodents and some birds (e.g., House Sparrows, *Passer domesticus*), may be abundant in built-up areas [70,71,72,73,74]. Conversely, in rural areas, the territory size was mostly negatively related to the seasonal progress and positively to altitude above sea level. The decrease in territory size during the season was also confirmed in the territory-level analysis. This pattern of seasonal changes in territory size corresponds to the findings of Finck [23] in Western Europe and reflects the annual reproductive cycle of this species. The Little Owl hunts in places where the height of vegetation is around 20 cm. During the whole breeding period, birds spent more than 85% of their time in areas where the vegetation was shortest [39,40,68,75]. The development of vegetation (May–July) covers the hunting grounds. Therefore, the area of territories is reduced. The larger territories of the Little Owl at higher altitudes may represent a gradient of habitat features or weather conditions changing with this variable [70,76]. Theory predicts that larger territories may provide less resources and lower survival probability than smaller territories [77]. In line with our findings in the Czech Republic and Biscay (Spain), areas where the Little Owl occurs are situated at lower altitudes [30,78]. This may have important consequences for conservation actions of this species. For example, nest boxes are more likely to be occupied by this species if located in areas of low (~100 m a.s.l) altitude [79]. In Bulgaria, most (~86%) of the Little Owl territories were located up to 300 m a.s.l. [80], thus, implementing conservation measures at higher altitudes would not be very beneficial. The sizes of the territory in both habitats were similar to those found in other studies. In Germany, the mean annual territory size was 12.3 ha [23]. The territory size in our study was the largest in early spring (breeding period), and similar findings were reported by Finck [23], Glue, and Scott [49].

### Study Limitations

Our study has several limitations that should be considered when interpreting the results. First, we did not measure building/structure availability (e.g., in random points) in the two landscape types. Availability of some structures such as poles or fences was very difficult during surveys. Thus, we were unable to analyze habitat selectivity and habitat preferences for territorial displays. Therefore, all differences in territory size and behavior probably mirror structural and compositional differences between the two landscape types. Our results should be interpreted as the description of habitat use rather than habitat selection. Second, our study was conducted during three consecutive years. Therefore, we could not control some potentially confounding variables (e.g., food availability). However, their potential impact is equally distributed between environments—each year, birds in both landscape types were observed. Third, in our experiments, we exposed males to high levels of aggression and stress by displaying calls of a competitor. This could mimic a strong intruder not affected by any display of the territory owner. We assumed that with increasing distance from the territory center, the stress and aggression of the territory owner would drop, which may not be entirely valid. It is also possible that vocalizing males know their real neighbors well and thus respond differentially to a foreign intruder (e.g., they might attack a dummy bird over a larger distance than known neighbors). Thus, the estimates of territory size may differ from real ones. However, as birds were treated in a similar way in both landscape types, the differences in territory size probably remain true.

## 5. Conclusions

Our study provides the first data on the differences in territorial aggressive behavior of the Little Owl between urban and rural landscapes. The rate of aggressive displays was similar between the two landscapes, and this suggests that, generally, territory defence in this species is not strongly affected by the density of urbanization [81,82,83,84,85,86]. However, we also demonstrated that in the two landscapes, birds differentially utilized various structures to display this behavior. Knowledge of the Little Owl’s territorial behavior is key to the active protection of this owl. The compositional differences between various territories may have important consequences for the behavior and reproductive success of birds [74,75,82]. Recent research confirms this—e.g., the great tits (*Parus major*) in urban areas were more aggressive than in rural areas [84]. Thus, the differences between urban and rural territories found in our study emphasize the need for further studies on the reproduction and behavior of the Little Owl to understand the demographic consequences of the successful colonization of urban landscapes by this species.

## Figures and Tables

**Figure 1 animals-14-00267-f001:**
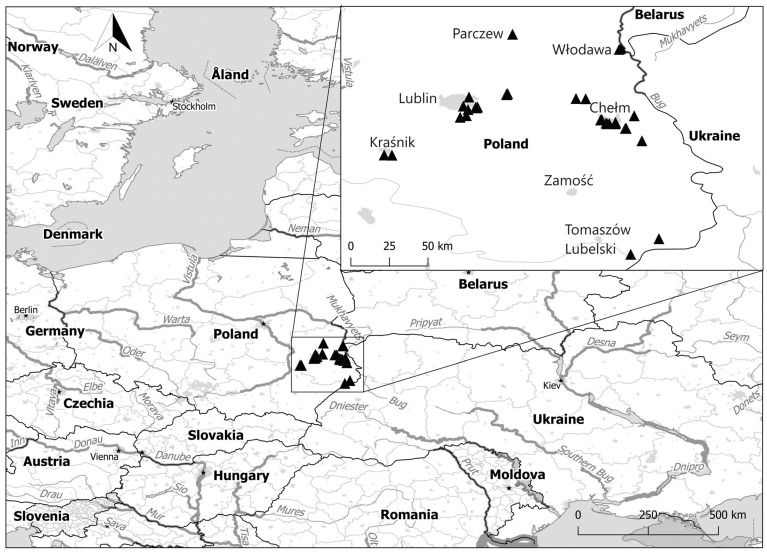
The study area (▲—observed territories of the Little Owl; ★—location capital of the country).

**Figure 2 animals-14-00267-f002:**
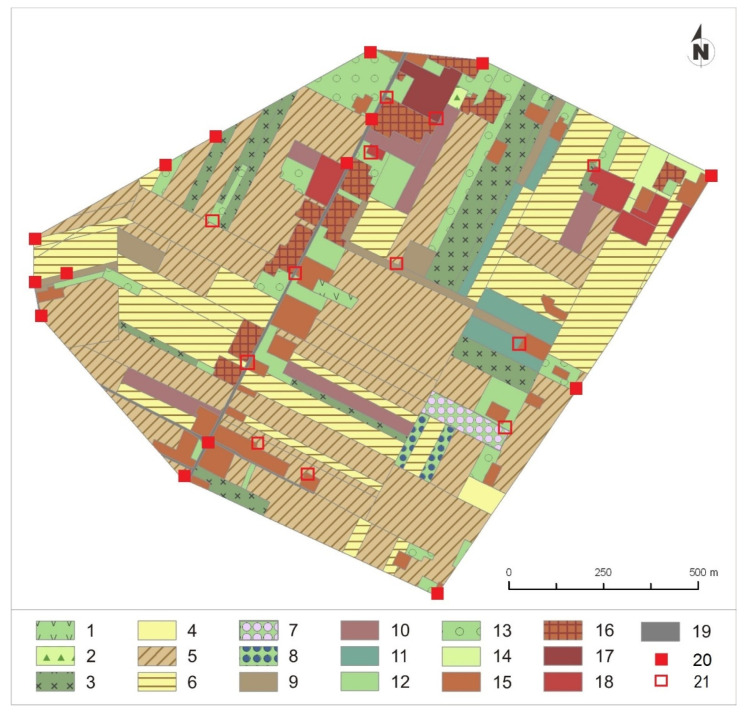
An example of the territory composition of the Little Owl in rural environments—a locality in Kalinówka near Lublin (geographical coordinates: 51.2019867778872, 22.660037372810823). Explanation: 1—pasture, 2—red clover, 3—fallow land, 4—corn, 5—wheat, 6—barley, 7—cuckooflower, 8—chokeberry, 9—potatoes, 10—sugar beets, 11—cabbage, 12—vegetable garden, 13—orchard, 14—grass, 15—rural buildings, 16—houses, 17—industrial buildings, 18—other buildings, 19—asphalt road, 20—attacks in this territory, 21—territorial call.

**Figure 3 animals-14-00267-f003:**
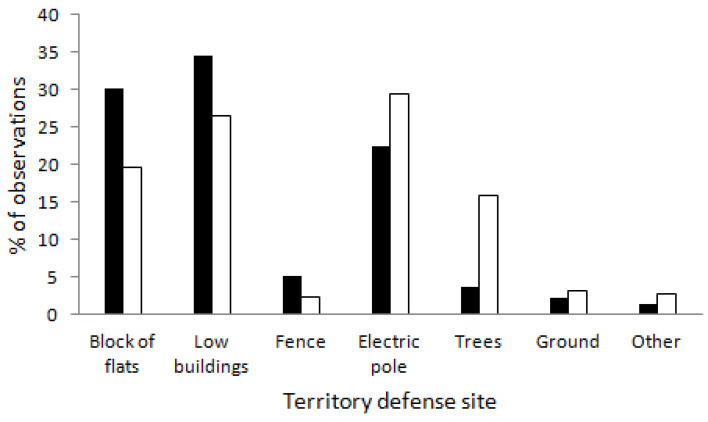
Differences in territory defence sites in urban (black bars) and rural (white bars) landscapes.

**Figure 4 animals-14-00267-f004:**
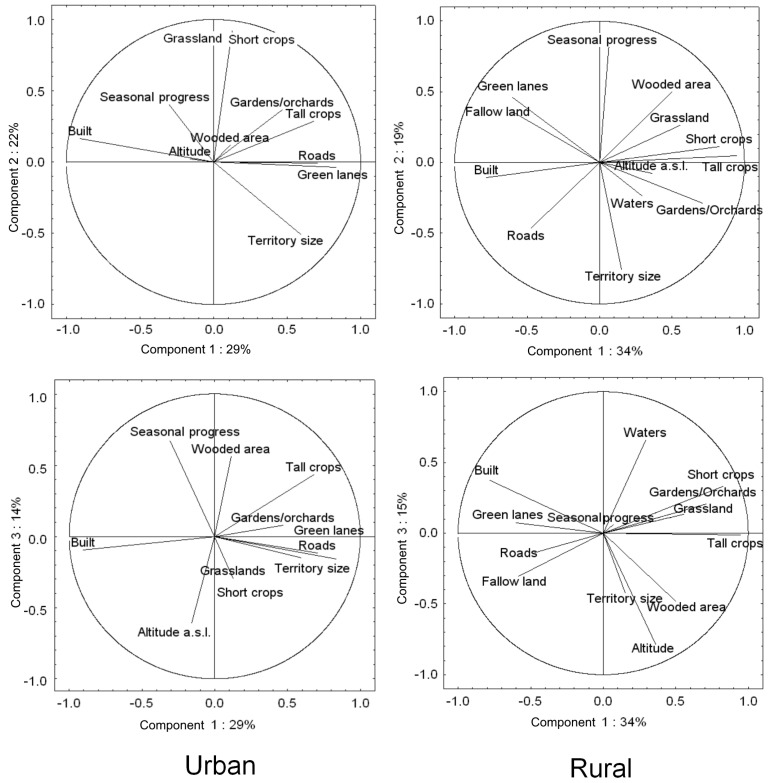
Results of PCA for the association between territory size and environmental variables in urban and rural landscapes. The first three principal components are displayed for each landscape type.

**Figure 5 animals-14-00267-f005:**
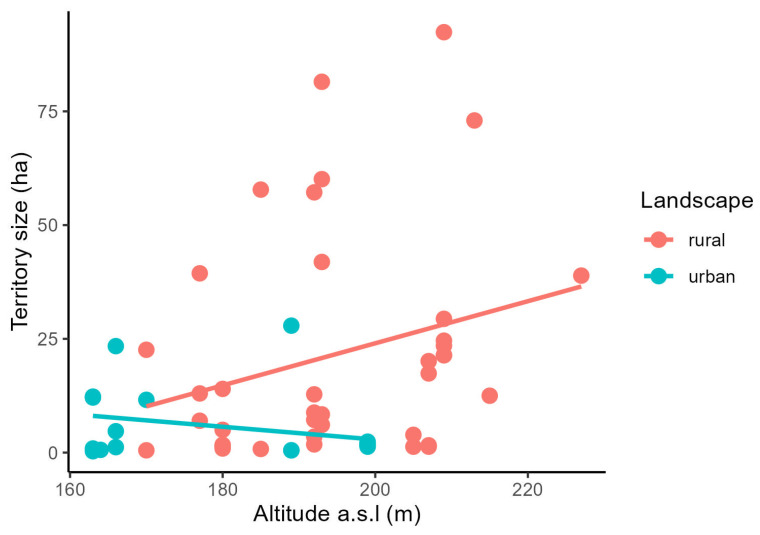
The association between the territory size of the Little Owl and altitude in rural (red) and urban (blue) landscapes.

**Table 1 animals-14-00267-t001:** Mean number of aggressive displays per one hour from various sites within the territories of the Little Owl in urban and rural landscapes. Statistically significant differences (based on *t*-tests) are emboldened.

Defence Site	Urban	Rural	*p*
	Mean ± SEM	Mean ± SEM	
**Block of flats**	5.6 ± 0.8	2.8 ± 0.8	**0.048**
**Low building**	4.2 ± 1.2	1.4 ± 0.5	**0.029**
Fence	0.8 ± 0.1	1.1 ± 0.4	0.508
**Electric pole**	2.1 ± 0.5	3.7 ± 0.6	**0.050**
Tree	1.2 ± 0.3	1.3 ± 0.2	0.830
Ground	0.7 ± 0.1	0.5 ± 0.1	0.208
Other	0.8 ± 0.3	0.5 ± 0.1	0.302
Total no. displays	7.5	8.7	0.603

**Table 2 animals-14-00267-t002:** Comparison of the habitat variables (percentage cover of land) within the territories of the Little Owl in urban and rural landscapes. Water and fallow land were not present within the territories of the Little Owl in the urban landscape. Statistically significant differences (based on *t*-tests) are emboldened.

Habitat Type	Urban (*n* = 17)		Rural (*n* = 16)		*p*
	Mean ± SEM	Min–Max	Mean ± SEM	Min–Max	
**Grassland**	0.0 ± 0.0	0–0.4	0.5 ± 0.2	0–5.2	**0.037**
**Tall crops**	1.7 ± 0.4	0–47.4	10.3 ± 0.6	0–84.6	**0.027**
**Short crops**	0.1 ± 0.1	0–1.3	2.3 ± 0.3	0–18.6	**0.002**
**Gardens and orchards**	0.4 ± 0.2	0–24	3 ± 0.4	0–60	**0.021**
**Built up areas**	65.6 ± 0.1	14.7–100	17.9 ± 0.4	0.3–100	**0.002**
Green lanes	4.1 ± 0.5	0–46.7	2.1 ± 0.4	0–20.5	0.325
Waters	-	-	0.1 ± 0.1	0–2.9	-
Roads	2.7 ± 0.3	0–13.3	1.6 ± 0.3	0–62.5	0.402
Wooded area	0.2 ± 0.2	0–9.8	0.7 ± 0.3	0–16.7	0.173
Fallow land	-	-	1.6 ± 0.4	0–20	-

**Table 3 animals-14-00267-t003:** Correlations between the habitat variables (including percentage covers) within the territories of the Little Owl and the first three principal components of the PCA in urban and rural landscapes.

	Urban			Rural		
	Component 1	Component 2	Component 3	Component 1	Component 2	Component 3
Territory size	0.591	−0.509	−0.152	0.149	−0.756	−0.421
Altitude a.s.l.	−0.158	0.021	−0.608	0.360	−0.076	−0.786
Seasonal progress	−0.302	0.403	0.664	0.067	0.895	0.058
Grassland	0.125	0.925	−0.294	0.554	0.263	0.132
Tall crops	0.680	0.286	0.435	0.946	0.046	−0.014
Short crops	0.125	0.925	−0.294	0.825	0.112	0.331
Gardens and orchards	0.467	0.365	0.079	0.710	−0.287	0.207
Built up areas	−0.905	0.164	−0.094	−0.782	−0.109	0.372
Green lanes	0.834	−0.037	−0.159	−0.602	0.461	0.075
Water	-	-	-	0.291	−0.236	0.657
Roads	0.704	−0.007	−0.115	−0.471	−0.463	−0.138
Wooded area	0.113	0.119	0.563	0.497	0.498	−0.479
Fallow land	-	-	-	−0.584	0.347	−0.307

## Data Availability

Data are contained within the article.

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
