# Peer review of "Little Owl Aggression and Territory in Urban and Rural Landscapes"

_animals, 2024, doi:10.3390/ani14020267_

Round 1

Reviewer 1 Report

Comments and Suggestions for Authors

Manuscript ID: animals-2721622

The aggressive behaviour and territory characteristics in the Little Owl Athene noctua: comparison between urbanized and rural landscapes

Grzegorz Grzywaczewski *, Federico Morelli, Piotr Skórka

Review

According authors, the aim was to study how differ some characteristics of breeding ecology of Little Owl between urban and rural landscapes, focusing mainly on: [1] territorial aggressive displays, [2] territory size and habitat composition within territories, and [3] factors that affect territory size.

Manuscript has a potential to be published, however, I have several comments to be addressed before acceptance.

General comments

1.       Paper do not fully conform to Template

2.       References should be formatted as per requirements of journal

3.       Reference [46] was not cited in the right order

4.       Latin names must be italicized

5.       Figures ant Tables must be cited consecutively

Title

Too long and repeats Keywords. Can it be shortened, like “Little Owl Aggression and Territory in Urban and Rural Landscapes”?

Simple summary is missing

Abstract

It is too long (200 words allowed), method not shown. Please do not include issues, not confirmed by results.

Material and Methods

Please include a map of the study area with study sites shown.

Please supply data on the altitude of study sites. Am I right that altitude in Lublin region is 100–300 m a.s.l.? Can such small differences have ecological consequences? See:

Šálek, M., & Schröpfer, L. (2008). Population decline of the little owl (Athene noctua Scop.) in the Czech Republic. Polish Journal of Ecology56(3), 527-534. – but no data on altitude given

Zabala, J., Zuberogoitia, I., Martínez-Climent, J. A., Martínez, J. E., Azkona, A., Hidalgo, S., & Iraeta, A. (2006). Occupancy and abundance of Little Owl (Athene noctua) in an intensively managed forest area in Biscay. Ornis Fennica83(3), 97. A.s.l. 0–1475 m, important but not explained

Gottschalk, T. K., Ekschmitt, K., & Wolters, V. (2011). Efficient placement of nest boxes for the little owl (Athene noctua). Journal of Raptor Research45(1), 1-14.

Ignatov, A., & Popgeorgiev, G. (2021). Recent and historical distribution of Little Owl (Athene noctua) in Bulgaria. Airo11, 216-222. - The elevation distribution reveals that most Bulgarian Little Owls are breeding up to about 300-400 meters asl. No more than 9% prefer higher altitudes from 400 to 800 m. Just 5% are above 900 m and several localities up to about 2300 m, mainly in mountain huts.

Line 118: some – how many, proportion of the total, etc.

Lines 120–124: Introduction or Discussion. Methods might start with “we modified above mentioned method of Finck, “

Table summarizing investigation effort would be great.

Line 141: wrong referring to Fig. 2, measurement of the height is not in the figure.

Line 163: mistype? If not, what are these forms 4 to 7?

Line 182: where above? Can you list variables here?

Line 184: differences of frequencies should be tested by G-test, not Chi-square.

Line 186: define abbreviation at the first use.

Results

Lines 200–203 – are these differences significant? Use G-test to compare proportions in pairs and in general, not Chi-square.

Line 208: mistype in df?

Table 1 – there is no plural of water in English.

Line 225-226: as already said, I doubt so small differences in a.s.l. can have significant impact. If authors stay on their own, then explanation in Discussion must be added, and I am sure, for such an unusual finding, simple graph a.s.l. – territory size must be added (possible as Supplement, but better in the main text).

Lines 228–232: explain with interpretation, what is this

Figure as example of the study site in urban environment must be added, similarly to Figure 1.

Figure 1 – use the same form, singular or plural, in the caption.

Figure 2 is not described in Results. Please also give some biological interpretation of these factors.

Table 1. For better readability, reduce number of columns, presenting mean±SEM, and min–max. How can df have a fraction?

Table 2. Please also give names and some biological interpretation of these factors.

Table 3. please present n, mean±SEM in separate columns; df is not needed.

By the way, testing calculation of Table 3 show

mean

SEM

mean

SEM

t

5.6

0.8

2.8

0.8

2.474874

4.2

1.2

1.4

0.5

2.153846

0.8

0.1

1.1

0.4

-0.72761

2.1

0.5

3.7

0.6

-2.04859

1.2

0.3

1.3

0.2

-0.27735

0.7

0.1

0.5

0.1

1.414214

0.8

0.3

0.5

0.1

0.948683

7.5

1.2

8.7

2

-0.5145

So, either give more digits (not recommended), or exclude also column t, to avoid questions.

Discussion

Lines 313–314: there are many references to what you say “may be “ – just a few examples below.

Lesiński, G., Gryz, J., Krauze-Gryz, D. et al. Population increase and synurbization of the yellow-necked mouse Apodemus flavicollis in some wooded areas of Warsaw agglomeration, Poland, in the years 1983–2018. Urban Ecosyst 24, 481–489 (2021). https://doi.org/10.1007/s11252-020-01046-7

Gortat, T., Barkowska, M., Tkowska, A. G. S., Pieniążek, A., Kozakiewicz, A., & Kozakiewicz, M. (2014). The effects of urbanization—small mammal communities in a gradient of human pressure in Warsaw city, Poland. Polish Journal of Ecology, 62(1), 163-172.

Gentili, S., Sigura, M., & Bonesi, L. (2014). Decreased small mammals species diversity and increased population abundance along a gradient of agricultural intensification. Hystrix, 25(1), 39-44.

Balčiauskas, L., & Balčiauskienė, L. (2020). On the doorstep, rodents in homesteads and kitchen gardens. Animals, 10(5), 856.

Line 325: check you references for [70,72,73,77,78]. Only one really gives a context of relation of owl territory size and altitude.

Lines 333–337: I failed to find data on availability of mentioned structures in two compared habitats. To have a conclusion that item is preferentially used, you must do selectivity analysis. Are there electric poles in urban environment?

Conclusions

This part is obviously too long and repeat other parts of the manuscript. Apart from being not supported by analysis (Lines 360–363, selectivity was not analysed), text need to be shortened and structured.

Back matter

Was Institutional Review Board Statement required for the study?

Comments on the Quality of English Language

Editing of languge required

Author Response

Please see details in the attachment.

Reviewer 2 Report

Comments and Suggestions for Authors

This manuscript deals with the very interesting idea of behavioral differences between urban and rural Little Owls in terms of territorial defense. In my opinion this is a very nice and inspiring idea and the Little Owl is a very suitable model species for that. Unfortunately the manuscript has a couple of flaws and will only reach publication stage after - in parts fundamental -  revision.

1. as in all studies dealing with selection decisions of animals , e.g. for habitat, food, or whatever, it is inevitable that selection is not measured in absolute numbers but always against the available options the animal had. Unfortunately, this basic rule is neglected in the manuscript. For example, when counting how often territorial calls are emitted from posts on buildings in habitat A and in habitat B it needs to be shown how many (suitable) buildings are available in A and in B. If in A twice as many buildings are used but at the same time A holds twice as many buildings there is random selection and the statement that twice as many buildings are used in A in comparison to B is without biological meaning.

2. the description of the methods does not allow to fully understand the results. It is not fully explained how aggressive behavior has been measured and how it was turned into numbers for statistical analysis. It seems to be the amount of calls per hour, but are all call types treated equally? The species has a quite wide spectrum of calls in different contexts. And was there no distinction between calling and attacking? In a paper with the title "The aggressive behavior...." I would surely expect that. Furthermore, in line 177 you mention "repeated surveys" without explaining how that was done (frequency, time lags in between, how often, tested for habituation effects?).

3. Furthermore, a discussion of the validity of the method is missing. I personally have strong doubts that pushing a male to high levels of aggression and stress by displaying  calls of a competitor with following movement of the artificial competitor away from the assumed center of the territory reveals a valid picture of the borders of the territory. By this way you first mimic a super strong intruder that is obviously not impressed by any display of the territory owner and then you lure the stressed bird step by step away from the center of the territory. Do you really expect that after a certain amount of meters stress, hormones and aggression of the territory owner suddenly drop to zero and the bird piecefully returns to the center of the territory? Maybe that luring towards the assumed edges of the territory was made in several sessions with clear breaks in between, but then this needs to be described in the methods section. Also it has been shown that most singing bird males know their neighbors pretty well by their songs and a completely foreign intruder might be attacked over a larger distance than the known neighbors. This means your results might have little to do with the usual day-to-day situation. All this needs to be regarded, justified and evaluated in a methods discussion. 

4. there is also an ethical issue to be regarded when a wild animal is confronted with an (artificial) aggressive intruder for up to 5.5 hours! This is heavily inducing stress to the bird, partly with a considerable duration. In most European countries it would be forbidden to stress a strictly protected bird species that way. In some European countries this study would even qualify as an animal experiment and require an ethical permit. Maybe in Poland this is not the case and the Little Owl has a lower protection status, but an ethical statement by the authors in my opinion is needed.

5. looking at the patchy distribution of usable habitat elements (including linear elements like roadsides)  the calculation of minimum convex polygons (MCPs) to reveal territory size is unsuitable. There is a risk of a systematic error when patchiness of the used habitat elements is basically different between urban and rural territories. One of the two types could have more "unusable" areas between the patches and then an MCP will reveal too large and unrealistic territory sizes. I strongly recommend to calculate more elaborate territory sizes based for example on Autocorrelated Kernel Density Estimations (AKDE) / utilization distributions (if sample size allows). 

Comments on the Quality of English Language

The English used in the manuscript is almost always easy to understand and pleasant to read, however there are a few wrong or at least strange phrase constructions and quite many articles and prepositions are missing. Since I expect a fairly extensive revision of the manuscript I will not list all the cases I came across and as a non-native speaker I might have missed a lot more.

Author Response

Please see details in the attachment.

Reviewer 3 Report

Comments and Suggestions for Authors

The manuscript is devoted to an interesting problem, in particular the study of the adaptive behavior of the Little owl in an urbanized landscape. The authors compare several characteristics, such as habitat structure, size of individual territories, and aggressive behavior in rural and urban landscapes.

This study is interesting. The methodology for studying the territorial and aggressive behavior of the Little owl and the size of individual areas is clearly described, the conclusions are formally confirmed by adequate statistical analysis.

However, there are some comments and questions.

1.   It is not clear from the manuscript in which years this study was carried out.

2.  The climate of the study area is described for the 20th century (see Kaszewski et al, 1999 [54]), while the average annual precipitation and temperature have changed at the present time. It is necessary to clarify the years of this study. If the study was carried out in recent years, it is necessary to provide current climate data.

3.  Was this a one-year study, or was data collected over several years?

4.  If field data is collected over a period of one year, then the conclusions may not be as clear-cut. Interannual variability may influence the behavior of the Little owl. As is known, many factors that remained outside the scope of this study (for example, fluctuations in trophic factors, etc.) can affect the behavior of owls in different years and be indirectly related to landscape features.

5.  It follows from the methodology description that the authors compare only two territories: one urban and one rural site each? If the research was carried out only at two sites, then this is not enough. In my opinion, multiple urban and rural sites for studying the territorial behavior of the little owl should have been included in the field study design to exclude random factors. 

6.   In my opinion, there is no point in considering the third component when describing PCA results.

There are technical remarks.

1. A reference to source [46] is given after reference to sources [47-50] (see lines 78-80 and 88). Needs to be corrected.

2. There is confusion regarding references to Figures 2 and 3 in the text. Needs to be corrected.

3.  Significant probabilities should be highlighted in bold in Tables 1 and 3

The manuscript must be corrected and re-reviewed

Author Response

Please see details in the attachment.

Round 2

Reviewer 1 Report

Comments and Suggestions for Authors

I appreciate authors work in revising manuscript. However, they do not stick to requirements of the journal in full, and there are mistypes in the text.

Comments on the Quality of English Language

Language is understandable, MDPI language servise will do the rest,

Author Response

Thank you for your work.

Reviewer 2 Report

Comments and Suggestions for Authors

In my opinion all my issues have been dealt with. Either by adding sentences to the manuscript and/or by explaining why the suggestions cannot be fulfilled. The problem of not having measured the structure availability in the habitats makes the study a bit less valuable than it could be, but on the other hand this is not an argument to not publish it at all. Therefore I am voting for publication after a final round of minor corrections. Here are two that I came across:

line 125: there is something wrong with the Finck (1990) citation: “the defendant area for food and nestling”. The phrase as it is given in the manuscript is not existent in the original Finck (1990) text, therefore it is not a citation and should not be in quotation marks.  In fact Finck explains and follows the definition of others, but without the quotation marks it should be ok to use it that way. However "defendant" must be "defended" and "nestling" must be "nesting".

inserted text in line 139/140:  delete "of" before "availability" and replace "urabn" with "urban".

Comments on the Quality of English Language

Also the newly inserted sentences in line 220 ff. contain some mistakes like missing articles or wrong grammar (e.g. "season progress" has to be either "seasonal progress" or "progress of season"). This can easily be amended by a final language check which would also be a good idea for the rest of the manuscript

Author Response

We modified according to your comments.

Reviewer 3 Report

Comments and Suggestions for Authors

I'm satisfied with the new version.

Author Response

Thank you for your work.